# KNOWLEDGELESS LANGUAGE MODELS: DECOUPLING LINGUISTIC COMPETENCE AND FACTUAL KNOWLEDGE

## ABSTRACT

Language models capture a broad spectrum of human knowledge due to being trained on large and diverse real-world datasets. However, this knowledge is not always necessary for linguistic tasks and can contribute to hallucinated outputs, as real-world knowledge is inherently dynamic and context-dependent. Such behavior limits their applicability in domains where factual precision is critical, such as healthcare and law. Moreover, LLMs trained on large text corpora inevitably inherit societal biases present in their sources. In this work, we introduce Knowledgeless LMs (KLLMs), a class of models intentionally pretrained to forgo memorization of entity-specific knowledge while retaining structural and semantic understanding of language. We present our approach for designing and training these models and evaluate them across a spectrum of downstream tasks, including language understanding, commonsense reasoning, and context-based factual benchmarks. Our results show that KLLMs achieve competitive or superior performance compared to fully parametric LLMs, particularly when provided with the relevant context, while substantially reducing reliance on memorized world knowledge. This leads to lower hallucination risks and improved calibration, with more reliable confidence estimates. Overall, KLLMs demonstrate that strong linguistic and reasoning capabilities can be maintained without extensive factual memorization, highlighting knowledgeless pretraining as a promising paradigm for building more efficient, faithful, and controllable language models.

## 1 INTRODUCTION

Large language models (LLMs) have demonstrated remarkable capabilities in encoding and leveraging a vast array of human knowledge (Petroni et al., 2019; Brown et al., 2020; Winata et al., 2021; Heinzerling & Inui, 2021; Cohen et al., 2023a; Pan et al., 2023). Much of this success is explained by parametric knowledge acquired during large-scale pretraining, including extensive and diverse factual and entity-specific information. While such broad knowledge representation materially advances their capabilities across diverse downstream tasks, it also predisposes these models to hallucination, generating plausible but ungrounded content (Maynez et al., 2020; Devaraj et al., 2022; Tam et al., 2023; Kaddour et al., 2023; Huang et al., 2024), a phenomenon particularly problematic in high-stakes domains such as medicine or law. Hallucinations may stem from data-related artifacts—including outdated, biased, or misinformed content (imitative falsehoods)—as well as inherent modeling limitations such as memorization of spurious correlations or long-tail facts. Recent work suggests that this challenge stems from the nature of current training and evaluation paradigms, combined with the sheer scale of factual knowledge, which makes it infeasible to encode exhaustively within the finite capacity of a model (Tauman Kalai et al., 2025; Xu et al., 2024).

Considerable effort has been devoted to devising techniques for updating the parametric knowledge of LLMs through targeted factual knowledge editing (Meng et al., 2022). However, these approaches are often difficult to sustain (Wang et al., 2024), may be imprecise (Yang et al., 2024), or introduce unintended side effects (Cohen et al., 2024a; Zhong et al., 2023). As LLMs are trained on static datasets with a fixed cut-off date, their parametric knowledge is ill-equipped to keep up with the rapid pace of novel facts and entities relating to current events. Moreover, LLMs inevitably inherit societal biases embedded in their training data, manifesting as skewed, oversimplified, or stereotyp-

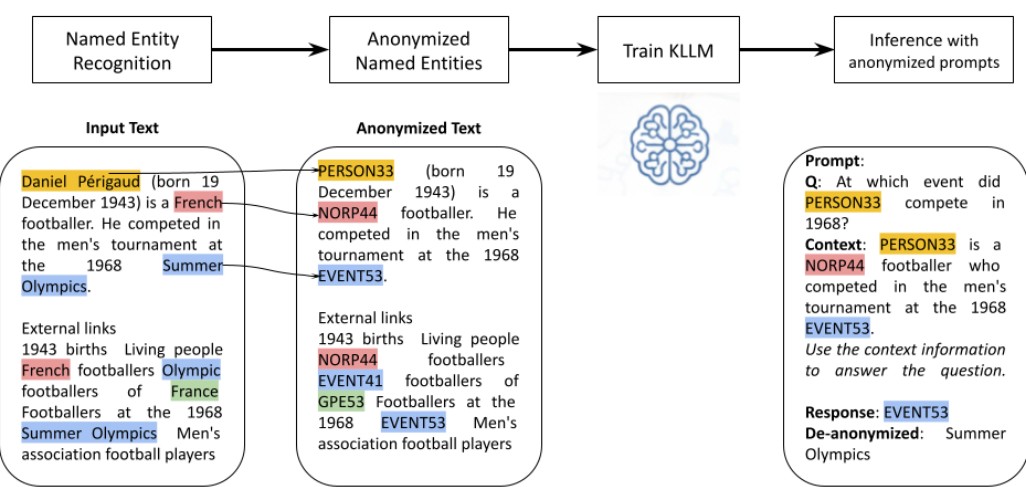

Figure 1: Pipeline of Knowledgeless Language Model (KLLM) training and inference

ical outputs (e.g., algorithmic bias, selection bias across demographics). The prevailing strategy to mitigate hallucination and account for new information involves grounding generation in external evidence, such as Retrieval-Augmented Generation (RAG) using a curated document store (Lewis et al., 2020) or graph (Edge et al., 2025). The LLM then no longer is required to maintain all pertinent knowledge in its weights. In this paper, we investigate the possibility of advancing this idea even further.

We introduce a novel class of models—KnowledgeLess Language Models (KLLMs)—that are intentionally trained to eschew specific world knowledge, in order to focus on the modeling of language while still retaining an abstract or structural understanding of how knowledge is organized and used. Beyond mitigating hallucination and bias, knowledgeless pretraining also offers practical advantages for the development and deployment of language models. Since KLLMs are not required to memorize vast amounts of entity-specific knowledge, they can be pretrained on smaller, less specialized corpora, thereby reducing computational costs and environmental impact. This lighter training burden makes the approach more sustainable and accessible, lowering barriers for research groups and organizations with limited resources. Moreover, by decoupling linguistic and structural competence from factual recall, KLLMs enable a sharper focus on task-specific adaptation: Models can be efficiently fine-tuned or contextualized for particular applications without carrying the overhead of redundant parametric knowledge. Finally, this separation makes KLLMs particularly well-suited for emerging agentic use cases—where models must act as adaptive, goal-driven systems that can ground their reasoning in validated context and dynamically adjust to user needs—an increasingly important direction in contemporary LLM research.

Hence, by overcoming the reliance on memorized facts, our goal is twofold: (1) to minimize hallucinations by reducing factual ambiguity, and (2) to lessen the amplification of societal biases found in real-world data. Surprisingly, our empirical evaluation finds that knowledge-light pretraining retains, and often strengthens, downstream performance. On SuperGLUE, KLLMs perform on par with or surpass baselines, particularly on reasoning-heavy tasks like COPA and WSC. When provided with the necessary context in factual reading benchmarks such as LAMA, SQuAD, and NQ, they consistently match or outperform baselines, showing that strong linguistic and reasoning abilities do not require memorized world knowledge. Closed-book evaluations further confirm that KLLMs carry substantially less factual memory, validating the success of anonymization. In addition, KLLMs exhibit improved calibration, with higher precision and more reliable confidence signals, making them better aligned for applications where trustworthy uncertainty estimates are critical. Together, these findings establish knowledgeless pretraining as a paradigm that holds potential for building more efficient, faithful, and controllable language models, particularly suited to domains where factual accuracy, interpretability, and safety are paramount.

Finally, to assess whether the effects of anonymization-based pretraining persist at larger scales, we additionally train KLLM–SLM pairs on a 10B-token subset of the SmolLM corpus (§4.6). These experiments confirm that the observed gains in contextual factuality and calibration remain stable with substantially more training data.

We further include extensive ablation studies on continued pretraining, inference-time anonymization, and anonymization robustness, together with qualitative analyses illustrating the behavioral differences between SLMs and KLLMs.

## 2 KNOWLEDGELESS LANGUAGE MODELLING

In order to develop an LLM that minimizes retention of real-world factual knowledge, we propose to preprocess pretraining data to markedly reduce the model's propensity to memorize specific factual knowledge from the data. In particular, we anonymize named entities by replacing their mentions with placeholder tokens, preventing the model from learning to associate any factual information with the names of entities such as people, places, or events. We train models from scratch on this preprocessed data without modifying the model architecture. During inference, the same anonymization procedure is followed to anonymize both the query and the context (where applicable), and entity names are restored as a postprocessing step. While some early work on reading comprehension (Hermann et al., 2015) employed a similar entity anonymization scheme during supervised training, we are not aware of previous work that applied this to pretraining and with the explicit aim of pretraining a model that lacks entity-specific factual knowledge.

### 2.1 CORPUS ANONYMIZATION

**Named Entity Recognition**   To effectively anonymize our pretraining data, we first seek to identify all named entities present in the text. We employ a state-of-the-art Named Entity Recognition (NER) model from the Flair framework (Akbik et al., 2019). In particular, we use the large 18-class OntoNotes model based on XLM-R embeddings,[1] which obtained a reported 90.0% F1 on OntoNotes (Schweter & Akbik, 2020).[2] We use OntoNotes' more fine-grained entity tagset (Hovy et al., 2006; Weischedel et al., 2011a) for better control over which kinds of entity mentions are anonymized and to be able to construct more informative placeholder tokens based on the entity types.[3] Importantly, in this work, we only anonymize tokens corresponding to named entity types, not numerical or temporal values, which are also recognized by the NER model. This is based on the conjecture that numerical and temporal values may be crucial for a deeper understanding of the textual context. Retaining them ensures that the anonymized text remains coherent, readable, and semantically meaningful, which is essential for effective model training without introducing bias.

Our OntoNotes-based NER tagger achieves an 87–90% mention-level F1 score, consistent with strong large-scale NER performance. This threshold offers an effective balance: more aggressive masking degrades syntactic integrity, while the residual unmasked 10–13% of entities still proves insufficient for meaningful factual retention. Indeed, closed-book QA accuracy remains near-random across all scales (§4.4, Table 5), empirically validating that this level of anonymization successfully suppresses parametric knowledge.

**Anonymization Strategy**   For each document in our dataset, we apply an anonymization procedure that replaces all identified entities with placeholders following the format ENTX, where ENT denotes the entity type and X is a unique identifier assigned within the specific document (see Appendix A). This strategy preserves the original sentence structure, enabling the model to discern general linguistic patterns and relationships without being biased by concrete real-world instances. By processing each document individually, we ensure that entity placeholders are unique within that document and consistent for repeated mentions of the same entity. This consistency helps maintain the coherence and logical flow of the text while still abstracting away factual details, thus preserving

---

[1] https://huggingface.co/flair/ner-english-ontonotes-large

[2] Based on a manual inspection of 50 randomly sampled documents from our pretraining corpus, we estimate an accuracy of approximately 87% in named entity recognition. In particular, rare or long-tail entity names appear less likely to be correctly identified.

[3] See Appendix A for a list of the detected entity types.

| | CNN/DailyMail | Wikipedia |
|---|---|---|
| Total # of tokens | 272M | 2.2B |
| Average # of tokens per sentence | 22.57 | 17.9 |
| Total # of articles | 300K | 7M |

Table 1: Pretraining corpus statistics comparing the CNN/DailyMail and Wikipedia datasets.

the underlying structure of general knowledge without exposing the model to specific facts. However, we do not perform coreference resolution and therefore different named mentioned of the same entity (e.g., "Barack Obama" and "President Obama") will generally be assigned different tokens. Figure 1 provides an example of text before and after applying our anonymization strategy.

This anonymization strategy notably limits models' ability to acquire specific factual information from the text, since most factual knowledge is grounded in one or more entity names. However, in addition to the fact that NER models do not have perfect recall (and therefore a small proportion of names will not be anonymized), some entities can be identified uniquely through descriptive references, which may cause some leakage of entity-specific knowledge. For example, the sentence *"The 44th President of the world's most powerful country was born on an island state in that country."* still encodes implicit entity knowledge, although our anonymization procedure was designed to remove such information. Another important limitation concerns gender information arising from the use of gendered coreferences. Referring to an entity with pronouns such as "he" or "she" implicitly reveals gender information and may lead the model to internalize gender-related biases. Nonetheless, we chose not to anonymize pronominal coreferences, as they constitute a fundamental linguistic component of language and communication that we want the model to capture.

## 2.2 PRETRAINING PROCEDURE

Knowledgeless language models are pretrained on large, diverse corpora similar to standard language models, the only difference being the application of the anonymization strategy described above that limits the model's direct exposure to entity-related knowledge. This ensures that the model learns general linguistic patterns while minimizing reliance on memorized entity-specific information. We train autoregressive Transformer language models with the standard language modeling objective of predicting the next token, but the approach is not limited to any particular model architecture. The KLLM's tokenizer should also be trained on the same anonymized corpus to ensure consistency. During fine-tuning and inference the same anomymization strategy is applied. This ensures that the model is learning how to gather information about entities and to reason about entities based on the given text without having memorized any information associated with entity names.

## 3 EXPERIMENTS

In the following, we introduce the details of experiments, including the considered models and baselines, training procedures, benchmark datasets, and evaluation protocols.[4]

**Pretraining Data** Our pretraining data consists of two English text corpora that complement each other in scope and content. The *CNN/DailyMail* dataset (Hermann et al., 2015) comprises thousands of English language news articles across diverse domains such as politics, business, sports, and technology, providing linguistically rich material with complex sentence structures and a high density of named entities, which makes it particularly valuable for contextual understanding. In contrast, the English *Wikipedia* (Bridge, 2001) offers comprehensive coverage of general world knowledge through structured expository passages. This can enable the model to acquire an underlying representation of the organization and structure of human knowledge, which can later be applied in downstream reasoning tasks. Table 1 summarizes the main statistics of the two corpora. *Wikipedia* is substantially larger in terms of token count and number of articles, and contains longer average sentences, whereas *CNN/DailyMail*, while smaller, provides denser entity mentions within its

---

[4]Our pretrained KLLMs, preprocessing and evaluation scripts will be released upon publication.

news-oriented domain. During preprocessing we remove non-UTF-8 characters, which reduces the number of redundant tokens in the tokenizer vocabulary. We concatenate these two corpora and anonymize entity mentions using the mechanism described in Section 2.

**Extended 10B-token Corpus.** To assess the scalability of anonymization-based pretraining, we additionally construct a 10B-token corpus derived from the publicly available `HuggingFaceTB/smollm-corpus`. This corpus is a high-quality mixture of CommonCrawl, Wikipedia, Books, News, and WebText-like sources, curated and sharded for large-scale pretraining. To ensure comparability with our 2.5B-token setup, we use the same preprocessing pipeline and sampling strategy: we uniformly sample text shards from the `smollm-corpus` mixture until reaching 10B tokens, preserving the dataset's original domain proportions. We do not apply additional filtering, deduplication, or weighting. The resulting dataset is therefore a larger but structurally matched version of our smaller corpus, enabling a controlled analysis of how anonymization interacts with increased data availability.

**Tokenization** We train BPE tokenizers (Sennrich et al., 2016) on our pretraining corpus. Anonymization tags (e.g. PERSON184) are added as reserved tokens in the tokenizer vocabulary to avoid them from being split into multiple tokens. Given that our training corpus is substantially smaller than the ones used for the original models considered in our study, we use a reduced vocabulary size of 30k tokens. For consistency we use the same vocabulary size for our baseline standard language models.

**Model Pretraining** We pretrain models from scratch using the LLaMA model architecture family (Touvron et al., 2023; Dubey et al., 2024). In particular we use the `Llama-3.2-1B` and `Llama-3.2-3B` models, as well as `SmolLM-135M`, `SmolLM-360M`, and `SmolLM-1.7B` from the HuggingFace SmolLM model (Allal et al., 2025). As baselines we train standard language models (referred to as SLMs) from scratch on our pretraining corpus but without applying the anonymization process. Therefore for each model architecture and size we can compare our knowledgeless language model (KLLM) to its SLM counterpart, differing only in its exposure to explicit entity information. Both models are trained on the same data size and for the same duration, which guarantees a fair comparison and allows us to attribute performance differences directly to the presence or absence of anonymization rather than other confounding factors. The pretraining loss curves are given in Appendix B. Additionally, we compare the results of both trained models against the original pretrained model snapshot, trained on orders of magnitude more data. This provides an external reference point to put our results in context.

**Fine-tuning** Since our pretraining is conducted at a comparatively small scale, we cannot assume strong zero-shot or few-shot capabilities. Therefore, for each evaluation benchmark we perform supervised fine-tuning using the corresponding training split of the dataset. To maintain consistency with our pretraining regime and to enable to the model to learn to make inferences based on anomyized text, the fine-tuning datasets are also anonymized prior to fine-tuning the knowledgeless models.

**Inference** Our inference pipeline follows a three-step anonymization process designed to align with the pretraining setup of our KLLMs. First, we anonymize both the input and its accompanying context using the same entity-masking scheme applied during pretraining. The anonymized data is then provided to the model for inference, ensuring that it operates without direct access to entity-specific information. Finally, the model outputs are *de-anonymized* by substituting back the original entities (see the example in Figure 1). This enables standard evaluation against the gold labels, ensuring that the evaluation faithfully reflects the intended knowledgeless setting while maintaining comparability to existing benchmarks.

## 4 RESULTS

To assess the performance of our KLLMs, we evaluate them on diverse downstream tasks in several different setups. Across all tasks, we follow the standard evaluation protocols defined in the respective benchmark setups, ensuring comparability with prior work. The main evaluation metric is accuracy, which is reported as a percentage.

|          | BoolQ | CB   | COPA | MultiRC | RTE  | WiC  | WSC  | Average |
|----------|-------|------|------|---------|------|------|------|---------|
| Original | 77.8  | 78.5 | 58.8 | 66.9    | 69.4 | 64.1 | 63.8 | 68.5    |
| SLM      | **70.6** | 75.0 | 55.7 | 64.4 | 62.8 | **61.2** | 61.4 | 64.4 |
| KLLM     | 70.5  | **77.9** | **59.0** | **64.5** | **66.5** | 58.8 | **63.7** | **65.8** |
| Δ        | −0.1  | 2.9  | 3.3  | 0.1     | 3.7  | −2.4 | 2.3  | 1.4     |

Table 2: SuperGLUE task results for models based on `Llama-3.2-1B`, including the original pretrained model, our standard model trained from scratch (SLM) and our knowledgeless model (KLLM). Accuracy is reported for all tasks except MultRC (F1).

| Model | LAMA | | SQuAD | | NQ | | FEVER | | HaluBench | |
|-------|------|------|-------|------|------|------|-------|------|-----------|------|
|       | SLM  | KLLM | SLM   | KLLM | SLM  | KLLM | SLM   | KLLM | SLM       | KLLM |
| SmolLM-135M (2.5B) | 15.8 | **18.6** | 15.2 | **16.1** | 4.0 | **4.1** | 82.8 | **89.5** | 58.5 | **61.2** |
| SmolLM-360M (2.5B) | 21.4 | **24.9** | 20.3 | **22.0** | 8.5 | **9.9** | 83.6 | **90.1** | 59.8 | **65.5** |
| SmolLM-1.7B (2.5B) | 43.2 | **48.5** | 55.9 | **59.8** | 16.3 | **19.2** | 86.9 | **94.7** | 64.3 | **74.7** |
| LLaMA-1B (2.5B) | 42.1 | **46.2** | 53.6 | **58.0** | 16.5 | **19.1** | 82.7 | **92.1** | 63.9 | **74.8** |
| LLaMA-3B (2.5B) | 46.8 | **49.8** | 50.5 | **53.9** | 22.1 | **26.9** | 87.5 | **94.9** | 67.7 | **75.8** |
| SmolLM-1.7B (10B) | 36.6 | **42.4** | 47.7 | **53.8** | 23.8 | **28.0** | 84.1 | **93.9** | 68.1 | **77.2** |

Table 3: Factual Reading accuracy results comparing KLLM and baseline across LAMA, SQuAD, NQ, FEVER, and HaluBench for different SmolLM model sizes and pretraining scales (2.5B and 10B tokens). For LAMA, SQuAD, NQ, and HaluBench we report accuracy; for FEVER, we report F1.

## 4.1 TASKS NOT REQUIRING ENTITY KNOWLEDGE

First, we employ the SuperGLUE benchmark (Wang et al., 2019). SuperGLUE is a suite of ten challenging language understanding tasks, covering areas such as question answering, textual entailment, co-reference resolution, and word sense disambiguation. The benchmark is particularly suitable for our evaluation, as its tasks are designed to test diverse aspects of language understanding and place a strong emphasis on reasoning. Table 2 presents the performance comparison between our KLLM, based on `Llama-3.2-1B`, against the baseline standard language model (SLM) across the SuperGLUE benchmark. The KLLM achieves comparable or slightly improved results on most tasks, with notable gains on CB (+3.9%), COPA (+5.9%), and WSC (+3.7%). Some tasks, such as BoolQ, RTE, and WiC, show small decreases relative to the baseline, with the largest relative drop observed on WiC (-3.9%). Importantly, these results demonstrate that, despite the pretraining preventing it from directly acquiring factual knowledge about entities, KLLMs still effectively capture the linguistic, grammatical, and semantic structure of the language, enabling strong performance on both reasoning and comprehension tasks. This suggests that parametric knowledge is not strictly necessary for maintaining a robust understanding of language structure and meaning, and that KLLMs can leverage task-specific contextual cues to achieve competitive performance. More broadly, these findings demonstrate that knowledgeless training may help reduce pretraining requirements when the downstream application does not require extensive world knowledge, making it a promising direction for developing models that are more efficient and easier to adapt to specialized domains.

## 4.2 FACTUAL READING COMPREHENSION

Our next set of experiments aims to assess whether our knowledgeless models are able to perform knowledge-intensive tasks including question answering, fact checking and hallucination detection when the necessary factual knowledge is provided as context. LAMA (Petroni et al., 2019) is designed to probe the factual knowledge encoded in language models by formulating cloze-style queries about world facts. SQuAD (Rajpurkar et al., 2016) is a large-scale question answering dataset based on Wikipedia passages, while Natural Questions (Kwiatkowski et al., 2019) provides real user queries paired with corresponding answers. Since our models are not expected to retain parametric factual knowledge, we adapt these benchmarks to a setup where the necessary background information is explicitly provided as context (as is already done in SQuAD). This allows us

| Model | CommonsenseQA | | StrategyQA | | PIQA | |
|---|---|---|---|---|---|---|
| | SLM | KLLM | SLM | KLLM | SLM | KLLM |
| SmolLM-135M (2.5B) | 26.5 | **31.1** | 52.2 | **55.5** | 65.9 | **69.6** |
| SmolLM-360M (2.5B) | 30.4 | **34.9** | 56.0 | **60.7** | 67.8 | **71.2** |
| SmolLM-1.7B (2.5B) | 34.6 | **39.6** | 61.8 | **66.7** | 70.5 | **76.2** |
| LLaMA-1B (2.5B) | 33.1 | **39.0** | 59.5 | **65.0** | 70.6 | **75.8** |
| LLaMA-3B (2.5B) | 40.8 | **45.5** | 65.0 | **69.7** | 72.3 | **78.5** |
| SmolLM-1.7B (10B) | 42.5 | **46.0** | 65.8 | **69.8** | 73.7 | **79.2** |

Table 4: Commonsense reasoning results comparing KLLM and baseline SLM accuracy across three benchmarks (CommonsenseQA, StrategyQA, and PIQA), for different SmolLM model sizes and pretraining scales (2.5B and 10B tokens).

to assess the models' ability to extract and reason over knowledge from context rather than relying on memorization.

FEVER (Thorne et al., 2018) is a large-scale fact verification benchmark, where claims must be supported or refuted using evidence from Wikipedia, directly testing a model's ability to ground predictions in verifiable context. HaluBench (Ravi et al., 2024) is designed to measure hallucination tendencies across diverse generation tasks, providing a fine-grained assessment of factual reliability.

Table 3 presents the performance of KLLM and SLM models across these benchmarks. On the factual reading tasks (LAMA, SQuAD, and NQ), KLLM training consistently improves performance over the baselines, with gains that grow larger at scale (e.g., +5.3 on LAMA and +3.9 on SQuAD with SmolLM-1.7B). These results confirm that removing entity-specific cues during pretraining does not weaken factual reasoning; rather, it encourages models to leverage the input context more effectively.

Strikingly, the improvements are even more pronounced on fact-checking and hallucination detection. On FEVER, KLLM models achieve up to +7.8 F1 over their baselines, while on HaluBench the gap reaches as high as +10.4% accuracy at the 1.7B scale. These benchmarks explicitly evaluate a model's ability to recognize misinformation and avoid generating unsupported claims, and the consistently higher scores of KLLM models suggest that knowledgeless pretraining strengthens their ability to abstain from or resist hallucinations. Together, these findings demonstrate that KLLMs not only retain robust factual reasoning under context but also provide an advantage in maintaining faithfulness and reliability.

## 4.3 COMMONSENSE REASONING

Furthermore, we evaluate our KLLMs on three widely-used commonsense reasoning benchmarks. CommonsenseQA (Talmor et al., 2019) tests the model's ability to answer multiple-choice questions that require broad everyday knowledge and commonsense inference. StrategyQA (Geva et al., 2021) challenges models to reason over implicit multi-step processes to answer yes/no questions, emphasizing reasoning rather than memorized facts. PIQA (Bisk et al., 2019) focuses on physical commonsense, assessing the model's understanding of everyday interactions and the principles of physical reality. In all cases, we provide the necessary context or task-specific information to the models, ensuring that the evaluation reflects their reasoning capability rather than reliance on parametric knowledge.

Table 4 summarizes the results of our SmolLM-based KLLMs and SLM baselines. KLLMs consistently outperform the baselines across all three tasks across all model scales, with improvements ranging from modest gains for smaller models to substantial margins for the 1.7B architecture (+5.0 on CommonsenseQA, +4.9 on StrategyQA, and +5.7 on PIQA). These results suggest that anonymization during pretraining does not impede the models' ability to capture commonsense patterns; instead, it encourages reliance on contextual reasoning rather than memorized associations. Combined with the reductions in hallucination observed in generation, this provides strong evidence that knowledgeless pretraining improves both factual robustness and commonsense generalization, particularly as the model capacity increases.

| Model | LAMA | | SQuAD | |
|---|---|---|---|---|
| | SLM | KLLM | SLM | KLLM |
| SmolLM-135M (2.5B) | 12.5 | 0.7 | 11.2 | 0.4 |
| SmolLM-360M (2.5B) | 20.8 | 1.2 | 18.6 | 0.9 |
| SmolLM-1.7B (2.5B) | 34.7 | 3.9 | 33.4 | 1.8 |
| LLaMA-1B (2.5B) | 34.5 | 3.7 | 33.1 | 1.7 |
| LLaMA-3B (2.5B) | 40.3 | 3.8 | 39.7 | 1.9 |
| SmolLM-1.7B (10B) | 27.3 | 2.1 | 24.7 | 1.1 |

Table 5: Closed-book QA accuracy results on LAMA and SQuAD comparing baseline (SLM) and KLLM performance across SmolLM and LLaMA models and pretraining scales (2.5B and 10B tokens). The consistently low results for KLLMs demonstrate that our approach successfully suppresses parametric factual recall.

| Model | Precision | | Recall | | F1 Score | |
|---|---|---|---|---|---|---|
| | SLM | KLLM | SLM | KLLM | SLM | KLLM |
| SmolLM-135M (2.5B) | 35.2 | **52.5** | 14.0 | **15.9** | 20.1 | **24.4** |
| SmolLM-360M (2.5B) | 36.4 | **55.7** | 19.8 | **21.5** | 25.6 | **31.1** |
| SmolLM-1.7B (2.5B) | 46.7 | **63.2** | 29.5 | **33.9** | 36.2 | **44.1** |
| LLaMA-1B (2.5B) | 43.9 | **59.6** | 29.1 | **32.1** | 34.9 | **41.9** |
| LLaMA-3B (2.5B) | 45.6 | **63.8** | 31.9 | **36.3** | 37.4 | **46.2** |
| SmolLM-1.7B (10B) | 48.5 | **63.7** | 29.2 | **34.4** | 36.5 | **44.7** |
| **Average (2.5B)** | 41.6 | **59.0** | 24.9 | **28.0** | 30.9 | **37.5** |

Table 6: Calibration evaluation reporting Precision, Recall, and F1 scores on the LAMA dataset when using the model's output probability to predict model correctness, for the SmolLM family, `Llama-3.2-1B`, and `Llama-3.2-3B`, comparing SLM and KLLM versions across pretraining scales (2.5B and 10B tokens).

## 4.4 CLOSED-BOOK QA

In order to assess that our knowledgeless models are not acquiring parametric knowledge despite anonymization, we additionally evaluate our KLLM models on the LAMA and SQuAD datasets in a closed-book setting, where no supporting context is provided. This enables us to quantify the effectiveness of our anonymization strategy, both because due to imperfect precision a small proportion of named entities are preserved, and to verify whether anomymization is sufficient to remove most factual knowledge.

Table 5 reports the results. The closed-book evaluation clearly shows that KLLM models retain only a minimal amount of factual knowledge compared to their non-anonymized baselines. While baseline models attain considerable accuracies, KLLM consistently lags far behind across all scales, with performance only marginally above random guessing. These findings demonstrate that KLLM training produces models that are largely knowledge-free, confirming that our anonymization strategy limits the accumulation of parametric knowledge by deliberately preventing exposure to entity-specific information during pretraining.

## 4.5 CALIBRATION AND FACTUALITY

**Calibration** We evaluate the calibration of each of our models when using the model's output probability to predict model correctness or certainty. A decision threshold is chosen through a parameter search on a held-out validation set, selecting the value that maximizes a model's own F1 score. The confidence signal used for this calibration is the probability assigned by the model to the first token in its generated output, which serves as a proxy for its certainty in producing a correct answer. Table 6 presents the results of each of our KLLM models compared to its corresponding baseline on the LAMA dataset. The evaluation metrics used here are precision (the proportion of

| Model | Precision | | Recall | | F1 Score | |
|---|---|---|---|---|---|---|
| | SLM | KLLM | SLM | KLLM | SLM | KLLM |
| SmolLM-135M (2.5B) | 50.4 | **69.8** | 8.0 | **10.5** | 13.9 | **18.0** |
| SmolLM-360M (2.5B) | 50.9 | **71.9** | 13.1 | **16.5** | 20.8 | **26.9** |
| SmolLM-1.7B (2.5B) | 51.5 | **74.4** | 26.4 | **29.3** | 34.8 | **42.0** |
| LLaMA-1B (2.5B) | 55.5 | **76.1** | 22.3 | **27.0** | 31.8 | **39.9** |
| LLaMA-3B (2.5B) | 55.1 | **75.8** | 29.9 | **30.7** | 38.8 | **43.7** |
| **Average** | 52.7 | **73.6** | 19.9 | **22.8** | 28.0 | **34.1** |

Table 7: Abstention tuning results, reporting Precision, Recall, and F1 scores on the LAMA dataset after abstention tuning on the SmolLM family, `Llama-3.2-1B`, and `Llama-3.2-3B`, comparing SLM and KLLM versions.

attempted predictions that are correct, recall (the proportion of correct answers the model attempts), and their harmonic mean, the F1 score, to capture trade-offs between correctness and coverage.

KLLMs consistently outperform the baselines across all model sizes and metrics. While the absolute recall values remain modest, KLLMs achieve considerably higher precision and balanced F1 scores, with gains becoming more pronounced as model capacity increases. On average, KLLMs surpass the baseline by +17.4 points in precision, +3.1 points in recall, and +6.6 points in F1. These results indicate that knowledge-light training improves the alignment between model confidence and correctness, enabling more reliable use of probability estimates as a calibration signal. Importantly, this suggests that removing parametric world knowledge may even enhance a model's ability to act as a calibrated predictor of its own correctness — an ability that is crucial in downstream applications where uncertainty estimation is central.

**Abstention Tuning** We additionally employ a factuality-oriented fine-tuning strategy aimed at encouraging models to abstain when uncertain rather than producing incorrect answers (Cohen et al., 2023a; Kadavath et al., 2022b). Specifically, we partition the training data into two subsets. The model is first fine-tuned in the standard way on the first subset. It is then evaluated on the second subset, and any instance where the model outputs an incorrect prediction is relabeled with the abstention output *"I don't know the answer"*. A second fine-tuning stage is then performed on this modified data, providing the model with explicit abstention supervision.

As shown in Table 7, this procedure yields consistent improvements in calibration and reliability. On average, KLLM models reach an F1 score of 29.0, compared to 23.2 for their baselines. Crucially, when compared against the standard pretrained LLaMA-8B—trained on orders of magnitude more data (trillions of tokens versus just 2.5B for KLLM)—the performance gap is remarkably small. For example, on SuperGLUE, our KLLM achieves an average score of 65.8, compared to 68.5 for the standard model, despite the latter's vastly larger training corpus and heavier pretraining regime.

These findings highlight two key insights: first, that knowledge-light models may be especially well-suited for abstaining from misinformation, since they rely less on memorized parametric knowledge and more on validated context; and second, that such models can remain highly competitive with standard LLMs even under drastically lighter pretraining.

### 4.6 SCALING TO 10B TOKENS

We now evaluate whether the effects of anonymization persist when substantially more pretraining text is available (§3.1). Using the same model architecture (SmolLM-1.7B), tokenizer, and optimization setup, we train SLM–KLLM pairs on the 10B-token corpus derived from the `smollm-corpus` mixture.

Tables 3, 4, 5, 6 show that the qualitative trends observed at 2.5B tokens remain stable: KLLM continues to outperform SLM on contextual factual QA, commonsense reasoning, and calibration benchmarks (measured as average F1), while preserving near-random closed-book accuracy (1.8). We attribute the slightly higher closed-book error at 10B tokens primarily to the domain shift in the `smollm-corpus`, which is substantially more out-of-distribution for closed-book QA than

Wikipedia-style data, making parametric factual recall intrinsically harder. Figure 4 depicts the corresponding training curves, confirming that both models converge and that the increased entropy introduced by anonymization again results in a slightly higher final loss despite improved downstream performance.

**Data Efficiency.** KLLM reaches the same contextual factual QA performance as SLM using approximately 38–45% fewer training tokens (KLLM@1.0B tokens $\approx$ SLM@2.5B tokens on FEVER and NQ), suggesting that removing entity-specific redundancy improves data efficiency in contextual reasoning.

Further ablations on continued pretraining, inference-time anonymization, and anonymization robustness, together with qualitative SLM–KLLM comparisons, are presented in Appendix D.

## 5 RELATED WORK

The primary motivation of this work is to develop models that are more robust by relying on externally provided, validated context rather than on their own parametric knowledge. As demonstrated both theoretically and empirically by Xu et al. (2024), hallucinations in LLMs are inevitable, since no model can fully encode the entirety of existing factual mappings. Additionally, the fact that traditional training and evaluation reward guessing more than uncertainty acknowledgment makes it natural for LLMs to hallucinate (Tauman Kalai et al., 2025). This topic has been studied from many different perspectives (Augenstein et al., 2023; Sahoo et al., 2024; Huang et al., 2025). This is also related to the setting of selective prediction, where models can abstain from answering a query (Varshney et al., 2022; Kamath et al., 2020).

Another complementary direction to reducing hallucinations is improving model calibration (Guo et al., 2017), i.e., aligning the model's confidence with the actual likelihood of correctness. This is particularly relevant to our work, as KLLMs are designed to abstain more readily from misinformation and benefit from mechanisms that quantify uncertainty. Prior approaches to calibration often operate at the logit level through post-hoc transformations (Desai & Durrett, 2020; Jiang et al., 2021), or rely on uncertainty estimation methods (Kuhn et al., 2023). More recent research has explored leveraging language models themselves for calibration, either by fine-tuning on correctness-labeled data (Kadavath et al., 2022a; Lin et al., 2022), prompting or in-context learning strategies (Cohen et al., 2023a; Alivanistos et al., 2022), or zero-shot instruction-oriented setups (Cohen et al., 2023b; Dhuliawala et al., 2023; Feng et al., 2024), as well as through consistency sampling (Yoran et al., 2023). Other approaches go further by exploiting internal representations for uncertainty classification (Azaria & Mitchell, 2023), introducing explicit tokens for abstention or uncertainty (Lu et al., 2022; Cohen et al., 2024b), or curating specialized datasets to train models to refuse unanswerable queries (Zhang et al., 2024; Cohen et al., 2025).

## 6 CONCLUSION

In this work, we introduced KnowledgeLess Language Models (KLLMs), a class of models deliberately trained to minimize reliance on parametric factual knowledge while retaining structural and linguistic understanding. Our experiments show that KLLMs maintain strong downstream performance, reduce hallucinations, and exhibit improved calibration, highlighting their reliability and suitability for high-stakes applications. By decoupling structural competence from memorized world knowledge, KLLMs offer practical advantages, including reduced pretraining costs, potentially lower environmental impact due to reduced pretraining requirements, and enhanced adaptability for task-specific fine-tuning. These properties make them particularly promising for agentic applications, where models must act as adaptive, goal-driven systems that ground reasoning in validated context and interact dynamically with users or external tools. Looking forward, RAG or other tool-use approaches could enable KLLMs to access high-fidelity knowledge dynamically, combining robustness against hallucination with factual grounding. Evaluating these models in real-world interactive settings will further reveal their capabilities, reliability, and alignment under open-ended deployment scenarios. Overall, KLLMs provide a promising pathway toward language models that are competitive, controllable, and practically deployable across diverse domains.

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

## A    Appendix: Anonymization Details

**Entity Recognition.**    Table 8 lists the entity types detected in the entity recognition phase, among which we only consider genuine named entities for anonymization, while retaining numeric values in the text.

| Named Entity Types | |
|---|---|
| **Entity Type** | **Description** |
| Person | People, including fictional |
| NoRP | Nationalities or religious or political groups |
| Facility | Buildings, airports, highways, bridges, etc. |
| Organization | Companies, agencies, institutions, etc. |
| GPE | Countries, cities, states |
| Location | Non-GPE locations, mountain ranges, bodies of water |
| Product | Vehicles, weapons, foods, etc. |
| Event | Named hurricanes, battles, wars, sports events, etc. |
| Work of Art | Titles of books, songs, etc. |
| Law | Named documents made into laws |
| Language | Any named language |
| **Number-Related Value Categories** | |
| **Value Type** | **Description** |
| Date | Absolute or relative dates or periods |
| Time | Times smaller than a day |
| Percent | Percentage (including "%") |
| Money | Monetary values, including unit |
| Quantity | Measurements, as of weight or distance |
| Ordinal | "first", "second" |
| Cardinal | Numerals that do not fall under another type |

Table 8: OntoNotes Entity Types (adapted from Weischedel et al. 2011b): Our anonymization strategy anonymizes named entities but not numerical values.

**Adding New Identification Tokens to the Vocabulary.**    In order to identify each of the specific entities within a document, we assign a specific identity token additionally to the entity type description (Section 2). Ideally, the generated tokens should exhibit maximal randomness to ensure minimal semantic overlap with any existing tokens in the language. To achieve this, we generate 100 novel tokens, each consisting of a randomly constructed string of 10 characters. Each character is sampled uniformly from a set comprising uppercase letters and digits. The choice of 100 tokens reflects a balance between distinctiveness—ensuring that each entity in a document can be uniquely identified—and frequency, such that each token occurs sufficiently often for the model's learned semantics to remain effectively random.

## B    Appendix: Pretraining Loss

For further analysis, in Figures 2 and 3, we plot the loss curves of our KLLM pretraining using `Llama-3.2-1B`.

During training, we observe that the non-anonymized baseline consistently achieves lower loss values compared to the anonymized models, with convergence occurring after less than one epoch. This behavior is expected, as explicit entity references provide stronger predictive cues than anonymized placeholders. For instance, predicting the continuation of *"Barack Obama was born in ..."* is considerably easier than for the anonymized variant *"Person74 was born in ..."*. The same holds for *"The iPhone was developed by ..."* in comparison with *"Product42 was developed by ..."*. A second factor contributing to this gap is that descriptive phrases referring to entities (e.g., *"the 44th President of the United States"*) are notably easier for the baseline model than for the knowledge-less version, which lacks entity-specific grounding. Finally, some loss discrepancies stem from limitations of our anonymization procedure itself, as the entity recognizer occasionally fails to anonymize certain

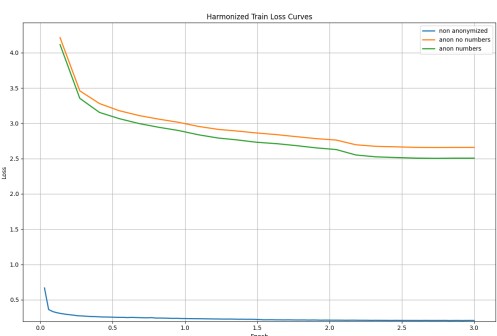

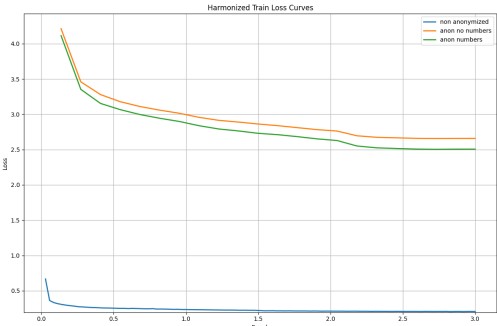

Figure 2: Training loss curves of the KLLM as well as of the baseline, for the `Llama-3.2-1B` architecture

Figure 3: Evaluation loss curves of the KLLM as well as of the baseline, for the `Llama-3.2-1B` architecture

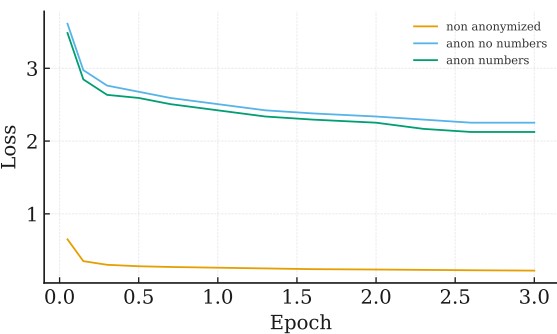

Figure 4: Training loss curves for SLM and KLLM pretrained on 10B tokens. Both models converge, and KLLM exhibits the characteristic higher entropy loss despite stronger downstream generalization.

mentions, leaving them as direct predictive cues. Together, these factors explain both the lower loss values and the earlier stagnation of the non-anonymized baseline.

## C   TRANSFER FROM ANONYMIZED TO NATURAL TEST DATA

KLLMs are trained and evaluated primarily under anonymized inputs, reflecting their intended operating regime where entity-specific surface forms are removed during pretraining. A fundamental question for practical deployment is whether the linguistic abstractions learned under anonymized supervision transfer robustly to **natural (de-anonymized) test inputs**. To directly assess this, we evaluate the **same KLLM model** under **open-book** conditions on both anonymized and natural versions of the test sets.

We focus on reasoning- and factuality-oriented benchmarks that require multi-hop, causal, or evidence-based reasoning: FEVER, CommonsenseQA (CSQA), StrategyQA, PIQA, and HaluBench.

### C.1 OPEN-BOOK QA: ANONYMIZED VS. NATURAL INPUTS (KLLM ONLY)

Table 9 reports KLLM performance under open-book evaluation on anonymized test inputs and on their corresponding de-anonymized natural counterparts.

| Task | KLLM (Anonymized) | KLLM (Natural) |
|------|------|------|
| FEVER | 93.8 | 89.6 |
| CSQA | 69.2 | 70.5 |
| StrategyQA | 61.4 | 60.4 |
| PIQA | 81.6 | 79.9 |
| HaluBench | 54.7 | 54.9 |

Table 9: KLLM performance under **open-book** evaluation on anonymized versus natural (de-anonymized) inputs.

| Model Variant | Factual QA | Commonsense | Closed-Book |
|------|------|------|------|
| SLM (from scratch) | 43.1 | 55.6 | 34.1 |
| KLLM (from scratch) | **48.1** | **60.8** | **2.9** |
| SLM $\rightarrow$ KLLM (cont.) | 46.8 | 58.9 | 28.5 |
| KLLM (no inference anonymization) | 89.6 | 70.3 | 2.9 |

Table 10: **Ablation Results.** Effects of continued pretraining and inference-time anonymization. Factual QA is averaged over LAMA, SQuAD, NQ, and FEVER; Commonsense over CommonsenseQA, StrategyQA, and PIQA; Closed-book over LAMA and SQuAD.

### C.2 STRUCTURAL TRANSFER ANALYSIS

Across all benchmarks, KLLM exhibits **stable performance under de-anonymization**, with only minor variations between anonymized and natural inputs. For knowledge-intensive verification (FEVER), performance decreases moderately (93.8 $\rightarrow$ 89.6), while for commonsense reasoning (CSQA) performance slightly increases (69.2 $\rightarrow$ 70.5). StrategyQA and PIQA show similarly small shifts, and HaluBench remains effectively unchanged.

These trends indicate that KLLM does not depend on surface-level entity placeholders for reasoning. Instead, its behavior is governed by **structural, relational, and semantic representations** that transfer robustly to real-world text. This confirms that anonymization-based pretraining does not introduce brittle reliance on artificial symbols and remains fully compatible with natural language inputs at inference time.

## D ABLATION STUDIES

**Continued Pretraining.** We evaluate whether the KLLM objective is effective only when training from scratch or also when applied on top of a standard pretrained model. To this end, we initialize from an SLM checkpoint and continue training under the KLLM anonymization objective. As shown in Table 10, continued pretraining improves contextual factual QA from 43.1 to 46.8 and commonsense reasoning from 55.6 to 58.9 relative to the SLM initialization. At the same time, closed-book accuracy drops from 34.1 to 28.5, indicating partial suppression of parametric factual recall even without full retraining from scratch. Crucially, full training under the KLLM objective further reduces closed-book accuracy to 2.9, confirming that parametric factual knowledge is maximally suppressed only when anonymization is applied throughout pretraining.

**Inference-Time Anonymization.** We also ablate the effect of removing anonymization at inference by evaluating KLLM on the de-anonymized (natural) test inputs. As shown in Table 10, factual QA on FEVER drops from 93.8 to 89.6 (–4.2 points), while the average commonsense score over CSQA, StrategyQA, and PIQA decreases slightly from 70.7 to 70.3. This modest degradation indicates that matching the train-time and test-time input distributions still benefits KLLMs, as test-time de-anonymization reintroduces surface-level entity forms that can re-activate weaker parametric priors and slightly misalign the model with the provided context.

**Anonymization Robustness: Masking Ratio.** We evaluate the sensitivity of KLLMs to the strength of anonymization by varying the fraction $p \in \{0.5, 0.75, 1.0\}$ of NER-identified entity

| Anonymization Regime | Factual QA | Commonsense | Closed-Book |
|---|---|---|---|
| SLM (no anonymization, $p = 0.0$) | 43.1 | 55.6 | 34.1 |
| KLLM ($p = 0.5$) | 44.5 | 55.9 | 20.9 |
| KLLM ($p = 0.75$) | 47.3 | 58.4 | 9.0 |
| KLLM (full anonymization, $p = 1.0$) | **48.1** | **60.8** | **2.9** |

Table 11: **Anonymization-strength ablation.** Impact of varying the fraction $p$ of NER-identified entity mentions that are anonymized during pretraining (SmolLM-1.7B, 2.5B tokens). Factual QA is averaged over LAMA, SQuAD, NQ, and FEVER; Commonsense over CommonsenseQA, StrategyQA, and PIQA; Closed-book over LAMA and SQuAD.

mentions that are replaced with placeholders during pretraining, while keeping the SmolLM-1.7B architecture and all optimization settings fixed.

Table 11 shows a clear monotonic trade-off as $p$ increases. Relative to the non-anonymized SLM ($p = 0.0$), partial anonymization already yields modest gains in contextual reasoning (Factual QA: $43.1 \rightarrow 44.5 \rightarrow 47.3$) while substantially reducing closed-book accuracy ($34.1 \rightarrow 20.9 \rightarrow 9.0$). Full anonymization ($p = 1.0$) further improves both factual QA (48.1) and commonsense reasoning (60.8), while driving closed-book performance down to near-random levels (2.9). These results indicate that increasing the anonymization ratio progressively suppresses parametric knowledge, with full anonymization providing the strongest separation between contextual reasoning ability and memorized facts.

**Qualitative SLM–KLLM Comparisons.** To provide a more concrete view of how KLLMs differ from standard language models in their reasoning behavior, we include a small set of qualitative examples drawn from FEVER, CommonsenseQA, StrategyQA, and HaluBench. In each case, the model is evaluated in the open-book setting with access to the relevant context.

| Dataset | Input (Natural vs. Anonymized; abridged) | SLM Output | KLLM Output |
|---|---|---|---|
| FEVER | **Natural:** "Taylor Swift wrote a song about artificial intelligence."
**Anonymized:** "PERSON46..7 wrote a song about artificial intelligence."
**Evidence:** No Wikipedia sentence directly supports or refutes this claim. | False | Not Enough Information |
| | **Natural:** "The actor who played Jack in Titanic was born in Los Angeles."
**Anonymized:** "The actor who played Jack in Titanic was born in GPE86..1."
**Evidence:** "Jack Dawson was portrayed by Leonardo DiCaprio." ("PERSON28..6 was portrayed by PERSON44..2."),
"Leonardo DiCaprio was born in Los Angeles, California." ("PERSON44..2 was born in GPE86..1") | Not Enough Information | True |
| Commonsense QA | **Natural:** "How do you show that you are agreeing with someone?"
**Anonymized:** "How do you show that you are agreeing with someone?" | Handshake | Nodding |
| StrategyQA | **Natural:** "Would a microwave heat up a brick faster than a glass of water?"
**Anonymized:** "Would a PRODUCT19..5 heat up a brick faster than a glass of water?" | Yes (69%) | No (61%) |
| HaluBench | **Natural:** "Who was the president of the United States during the Great San Francisco Earthquake of 1925?", "The president at the time was Calvin Coolidge."
**Anonymized:** "Who was the president of the GPE13..1 during the EVENT19..5 of 1925?", "The president at the time was PERSON86..1."
**Evidence:** "The Great San Francisco Earthquake occurred in 1906, not 1925." ("EVENT19..5 occurred in 1906, not 1925."),
"There was no major San Francisco earthquake in 1925." ("There was no EVENT19..5 in 1925.")
"Calvin Coolidge was president from 1923–1929, but not during the 1906 earthquake." ("PERSON86..1 was president from 1923–1929, but not during the 1906 earthquake.") | Hallucinated | Hallucinated |

Table 12: Qualitative comparison of SLM and KLLM predictions under open-book evaluation. Each example corresponds to the *same underlying dataset instance*, shown in both its original natural form (used for SLM evaluation) and its anonymized form (used for KLLM evaluation). The examples highlight failure modes of standard models such as hallucination and over-reliance on parametric priors, versus stronger evidence grounding and calibrated abstention in KLLMs.

