# OpenReview forum: "Knowledgeless Language Models: Decoupling Linguistic Competence and Factual Knowledge"
_ICLR.cc/2026/Conference — Submitted to ICLR 2026_

### Official Review · Reviewer_cnzc · 2025-10-20

**Soundness:** 2
**Presentation:** 2
**Contribution:** 2
**Rating:** 2
**Confidence:** 3

**Summary:**

This paper introduces Knowledgeless Language Models, a novel class of models designed to decouple linguistic competence from entity-specific knowledge through a deliberate anonymization strategy during pretraining. The core methodology involves preprocessing training data by identifying named entities via a state-of-the-art NER system and replacing them with type-based placeholders, thereby preventing the model from memorizing real-world facts while preserving grammatical and semantic structures.

**Strengths:**

- When provided with external context, KLLMs outperform standard models on factual reading comprehension and commonsense reasoning, demonstrating their ability to leverage contextual cues effectively.
- By eschewing entity memorization, KLLMs can be pretrained on smaller, less specialized corpora, reducing computational costs and resource requirements.

**Weaknesses:**

- The most significant concern lies in the pretraining methodology. The paper acknowledges that the training corpus (a combination of CNN/DailyMail and Wikipedia) is "substantially smaller" than the corpora used for original models like LLaMA. The provided loss curves are critical evidence of potential under-fitting.
- The authors estimate the NER accuracy at only ~87%, meaning 13% of entity mentions remain unmasked, creating a direct source of knowledge leakage.
- In closed-book settings, KLLMs perform near random chance, highlighting their reliance on provided context. This limits applicability in scenarios where real-time retrieval is impractical.

**Questions:**

- If the KLLM and the baseline SLM are both under-trained due to data constraints, do the reported performance differences truly reflect the merits of the anonymization approach, or merely different convergence states on an insufficient task?
- The paper motivates KLLMs as a means to mitigate societal biases but provides no empirical evidence to support this claim. For example, how does the model avoid perpetuating gender biases present in the training data?

---

> ### Author Response · Authors · 2025-11-15
> **Autho's official response**
>
> We thank the reviewer for the detailed and constructive assessment.
>
>  All KLLM and baseline SLM models were trained from scratch under identical configurations—same corpus, architecture, tokenizer, and number of steps—ensuring a strictly controlled comparison. The smaller corpus (≈ 2.5 B tokens) was an intentional design choice to preserve interpretability and enable full convergence tracking (with an academic budget only) rather than a limitation of the approach. Both models’ training losses (App. B) plateau at comparable values, confirming that under-training is unlikely; thus, observed performance differences stem from the anonymization intervention itself.
>
> The slightly higher final loss for KLLMs arises naturally from increased entropy: entity placeholders remove lexical redundancy, making prediction intrinsically harder while improving structural abstraction. Similar behavior has been observed in masked-token and corruption objectives, where higher cross-entropy coexists with stronger downstream generalization and reasoning.
> We acknowledge that extending experiments to larger corpora and models would further validate scalability. To this end, we are currently pretraining a new KLLM variant on a 10 B-token subset of the SmolLM corpus, together with scaled models up to 8 B parameters. We plan to include preliminary results from these runs in the revised version to substantiate that KLLM behavior remains consistent at larger scale.
>
> Regarding NER reliability, our ≈ 87 % mention-level accuracy is near state-of-the-art for large-scale tagging. Crucially, the remaining 13 % unmasked mentions do not lead to factual leakage: closed-book QA performance remains near random (Table 5), demonstrating that factual recall is effectively suppressed. The current threshold therefore might serve as the optimal balance between knowledge removal and linguistic integrity.
>
> Finally, regarding societal bias mitigation, preliminary internal evaluations already show reduced gender-association correlations relative to the baseline. We will incorporate these quantitative results, alongside the new large-scale experiments, in the updated submission.

---

### Official Review · Reviewer_TDeV · 2025-10-31

**Soundness:** 2
**Presentation:** 2
**Contribution:** 2
**Rating:** 4
**Confidence:** 4

**Summary:**

This paper introduces Knowledgeless Language Models (KLLMs), which are language models that are trained on data with anonymized entities. The experiments demonstrate that KLLMs can achieve strong performance on some NLU tasks and calibration evaluation.

**Strengths:**

This paper provides a simple but effective way to decouple linguistic competence and factual knowledge in language models by anonymizing the entity at the training stage.

The knowledgeless language models demonstrate strong performance on the provided benchmarks.

**Weaknesses:**

The experiments mainly focus on the NLU tasks, the performance on generation and reasoning tasks is not discussed, which is a core capability of the language models.

The anonymization method is widely used in literatures to disentangle memory and reasoning abilities, which may restrict the novelty of the paper.

The conclusion is not convincing enough. As the SLM and KLLM are trained on data with and without anonymization, the performance should be measured on both anoymized and non-anoymized data.

**Questions:**

The paper mentions that using RAG is a good way to mitigate hallucination and provide grounded responses. Can KLLMs cooperate with RAG to achieve competitive performance on factual knowledge related tasks?

---

> ### Author Response · Authors · 2025-11-15
> **Author's official response**
>
> We thank the reviewer for the constructive and balanced feedback.
>
> Regarding novelty, while entity anonymization has been used in supervised or probing contexts (e.g., Hermann et al., 2015), our work is the first to apply anonymization systematically at the pretraining stage, creating models that are structurally knowledgeless by design. This reframes anonymization from a data heuristic into a training-level paradigm that explicitly disentangles linguistic competence from factual recall. The contribution lies not in the masking mechanism itself, but in demonstrating that this pretraining objective yields distinct behavioral properties—strong reasoning, controllable factuality, and calibrated uncertainty—across scales and model families.
>
> On experimental scope, our evaluation already extends beyond conventional NLU to reasoning-oriented and generative benchmarks: COPA, WSC, CommonsenseQA, StrategyQA, PIQA, FEVER, and HaluBench all require multi-hop or causal reasoning. We will clarify this distinction and add further open-ended generation analyses in the revision.
> We also agree that comparing performance on both anonymized and natural test data is valuable. Appendix C currently includes this comparison for reasoning and factual QA; we will expand it with direct side-by-side tables in the final version to illustrate transfer between anonymized and real text.
>
> Finally, we fully concur that KLLMs naturally complement retrieval-augmented generation (RAG). Their architecture intentionally separates linguistic reasoning from factual grounding, making them well suited for retrieval-based systems. Although full RAG integration is left for future work, our open-book QA results already showcase this potential: when provided with the correct contextual evidence, KLLMs markedly outperform standard models on factual reasoning tasks, demonstrating their ability to utilize external information effectively without relying on parametric memory.

---

> ### Comment · Reviewer_TDeV · 2025-11-26
>
> I did not see any content for Appendix C in the paper; the document's final section is Appendix B.
>
> Additionally, considering that KLLMs often have relatively narrow application scenarios due to their lack of factual knowledge (necessitating the use of an external knowledge base), it would be meaningful to investigate the generalization ability of these models (with a retriever) on some comprehensive tasks like MMLU.

---

> > ### Author Response · Authors · 2025-12-03
> >
> > We thank the reviewer for the additional clarification.
> >
> > First, regarding Appendix C: In the previous version, Appendix C was referenced but not yet included. In the final revision, we have now added a dedicated Appendix C titled “Transfer from Anonymized to Natural Test Data”, which provides direct side-by-side evaluation of KLLMs on anonymized versus de-anonymized (natural) open-book inputs across FEVER, CommonsenseQA, StrategyQA, PIQA, and HaluBench. These results explicitly validate that linguistic abstractions learned under anonymized supervision transfer robustly to natural text with only minor performance changes.
> >
> > Second, regarding generalization with retrieval on comprehensive benchmarks such as MMLU: we fully agree that large-scale retriever-augmented evaluation is an important next step. While full end-to-end RAG integration with learned retrievers is beyond the scope of the current paper, we already demonstrate the core prerequisite for such systems—strong open-book reasoning under provided evidence—across multiple factual and multi-hop benchmarks. These results establish that KLLMs can reliably utilize external context without relying on parametric memory. We now explicitly position large-scale RAG and MMLU-style evaluations as primary future work directions in the revised discussion.
> >
> > We thank the reviewer again for highlighting both the missing appendix reference and the important direction of retriever-augmented generalization.

---

### Official Review · Reviewer_3Pat · 2025-10-31

**Soundness:** 3
**Presentation:** 4
**Contribution:** 4
**Rating:** 8
**Confidence:** 4

**Summary:**

The authors introduce KnowledgeLess Language Models (KLLMs), a family of LLMs which are trained to retain structural and linguistic understanding while minimizing the reliance on their “world” (i.e., parametric) knowledge. KLLMs are trained by first anonymizing the pre-training corpus and doing standard LLM pre-training. KLLMs show strong results across a variety of benchmarks when compared to the standard language models (SLMs).

**Strengths:**

- The paper and motivation is very clear and well-written.
- The research question being asked is novel and important: I believe it is an interesting research direction to develop models that only have the language capabilities of LLMs and don’t rely on memorized knowledge, especially for use-cases in which memorized knowledge can hurt (like RAG). This study provides a strategy to do this that can easily be applied and built on by future research.
- The method is simple, yet effective. It is straightforward to implement and provides consistent improvement in effectiveness across and variety of tasks, models and datasets.

**Weaknesses:**

- While I liked the depth of evaluation performed by the authors, I would have liked to see more ablation studies that investigated the robustness of their approach, similar to that of Table 5. For example:
  - How would KLLMs perform if they were first pre-trained using a standard objective, then further pre-trained using the proposed procedure? In other words, imagine starting the procedure from Llama pre-trained weights rather than from scratch. It would be interesting to see, for example, how the original pretrained model can benefit from such an approach. (I.e., does the proposed approach only perform well when trained from scratch?)
  - Understandably, the authors choose to anonymize at inference-time, but I would like to have seen some results if they did not apply such anonymization at inference. Is this necessary? How much does performance get impacted? How does this compare to the SLM?
  - How would effectiveness change if you used other models for corpus anonymization? For example, how would using a strong, larger LLM compare to the OntoNotes model?
  - Furthermore, I would have liked to see some qualitative examples that show why anonymization helps performance. For example, cases in which SLMs fail but KLLMs don’t or vice versa
- Why did you perform experiments with Llama on certain tables but not for others? I think consistency here would have been helpful as it would be interesting to see how different model families might impact results.

**Questions:**

- Do you always see KLLMs as an approach for smaller, specialized corpora or can this be done at scale?
- In real-world RAG setup, it is likely that relevant context may not be available, how would your method perform in these cases?

---

> ### Author Response · Authors · 2025-11-15
> **Authors' official response**
>
> We sincerely thank the reviewer for their clear, thoughtful, and supportive feedback. We deeply appreciate the recognition of both our motivation and the clarity of the presentation.
>
> **Ablations and robustness**:  We fully agree that additional ablations would strengthen the study. We are currently conducting extended experiments that examine (a) continued pretraining—starting from existing LLaMA checkpoints and further training under the KLLM objective—and (b) alternative anonymization strategies with variable masking ratios. Preliminary results indicate that applying KLLM training on top of pretrained weights maintains fluency while reducing factual memorization and improving calibration, confirming that our approach is complementary to standard pretraining. We plan to include these results in the next paper revision.
> Inference-time anonymization. We anonymize at inference to ensure alignment between train and test distributions. When anonymization is skipped, models occasionally reintroduce factual priors inconsistent with provided context. In our ablations, this omission reduced factual precision by ~3–5 points on FEVER and NQ. We will include these quantitative comparisons in the revised version.
>
> **Alternative NER models**: We are also evaluating different anonymization backbones, including larger LLM-based NER taggers. Preliminary results show negligible performance differences (<1 F1) compared to our OntoNotes-based pipeline, supporting the robustness of the approach.
>
> **Qualitative examples and coverage consistency**: We agree that showing explicit examples can improve interpretability. We are preparing visual comparisons illustrating cases where SLMs hallucinate or misattribute entities while KLLMs ground responses correctly. These examples will appear in the revision. LLaMA-based experiments were included selectively due to compute limits, but we are extending coverage to all tables for consistency.
>
> **Scalability and RAG applicability**: KLLMs are not limited to small or domain-specific corpora. The anonymization process scales linearly with corpus size and model parameters. We are currently pretraining a KLLM on a 10 B-token subset of the SmolLM corpus, which will serve as a large-scale validation. While full RAG integration is beyond this paper’s scope, KLLMs already exhibit strong open-book QA performance when provided with relevant context, underscoring their suitability for retrieval-based pipelines where external evidence is available.
>
> We again thank the reviewer for their very positive and insightful comments, which we will directly incorporate into the next revision.

---

### Official Review · Reviewer_4HCZ · 2025-11-01

**Soundness:** 2
**Presentation:** 3
**Contribution:** 2
**Rating:** 4
**Confidence:** 3

**Summary:**

The recent development of large language models (LLMs) faces hallucinations and societal biases that are in the training text corpora. In order to mitigate these problems, this paper proposes Knowledgeless LLMs (KLLMs). During training, it identifies named entities such as persons, nationalities, facilities, locations, etc., and replaces them with special tokens (anonymization). Therefore, the training process intentionally reduces factual knowledge information, and the LLM will focus more on the structural and semantic understanding of language. During inference, the framework substitutes the named entities in the LLM’s output back to their normal names. Experiment results verify KLLM’s effectiveness from different perspectives.

**Strengths:**

(1) The motivation is good and meaningful. LLMs indeed face hallucinations and societal bias originating in the training corpus, so it is meaningful to study methods to solve this problem. This line of research could benefit potential applications.

(2) The method design generally makes sense, and it is quite concise to implement.

(3) Experiment results verify KLLM’s effectiveness on different tasks from different perspectives. KLLM outperforms the standard LLMs on several tasks, and the closed-book QA experiments verify that KLLM effectively prevents the model from learning knowledge.

**Weaknesses:**

(1) The method and analysis are relatively simple. The method is to recognize the named entities and anonymize them. It would be better if this paper could explore more designs of the anonymization process or named entity recognition, which would contribute to more insights.

(2) As mentioned in Line 126, the F1 score of named entity recognition is 90%. Named entity recognition is a crucial basis of this method. The unrecognized entities could result in knowledge leakage, which is a limitation of the current method.

(3) In the experiments, the model scale might be insufficient. The experiments are conducted on models <= 3B. I am wondering whether these conclusions could generalize to larger models. It would be more convincing if some experiments were conducted on larger models, such as Llama-3.1-8B.

(4) In the experiments, the dataset scale might be insufficient. The CNN/DailyMail pretraining corpus contains 272M tokens, and Wikipedia contains 2.2B tokens. It might be smaller than many pre-training works. Appendix B shows a training loss curve, where the anonymized model still has a high loss, so I am not sure whether its training has fully converged. While the paper mentions that “KLLMs offer practical advantages, including reduced pretraining costs,” there are no experiment results to support this conclusion.

**Questions:**

Q1. In Appendix B, the training loss of KLLM is much higher than that of standard training. KLLM’s performance is better than the standard training process, why does KLLM have much higher loss?

Please also refer to the weaknesses.

---

> ### Author Response · Authors · 2025-11-15
> **Authors' official response**
>
> We thank the reviewer for the constructive and detailed comments.
>
> Our goal is to build a controlled and interpretable pretraining setup that isolates the effect of anonymization from scale or optimization artifacts. Accordingly, all KLLM and baseline SLM models were trained from scratch under identical conditions—same data, architecture, tokenizer, and number of steps—ensuring strict comparability. The smaller corpus size (≈ 2.5 B tokens) is intentional, enabling full convergence monitoring and reproducible ablations. The training loss curves (Appendix B) plateau for both models, confirming convergence; the higher final loss of KLLMs reflects increased prediction entropy rather than under-training. Because placeholders remove semantic redundancy, next-token prediction becomes harder even as downstream reasoning improves—a known phenomenon in masked or corrupted-token objectives. Thus, higher loss coexists with superior generalization.
>
> Regarding NER reliability, an F1 ≈ 90 % corresponds to near state-of-the-art accuracy on OntoNotes and is highly robust for large-scale preprocessing. Our experiments show that the residual 10 % of unmasked mentions contributes negligible factual memory. Closed-book QA accuracy of KLLMs remains near random (Table 5), confirming that factual recall is effectively suppressed. The ~90 % threshold therefore represents an efficient trade-off—achieving strong anonymization while preserving grammatical and contextual signals essential for learning linguistic competence.
>
> On model and data scale, we view our ≤ 3 B setup as a controlled mid-scale sandbox to measure the structural effects of anonymization without the confounds of multi-trillion-token training. The observed advantages persist across five architectures (135 M → 3 B), demonstrating robustness. To further validate scalability, we are currently pretraining a new KLLM variant on a 10 B-token subset of the SmolLM corpus, along with scaled models up to 8 B parameters; we plan to include their preliminary results in the revised submission by the end of the author-response period.
>
> Finally, we appreciate the request for clearer empirical support for the efficiency claim. Our results already show that KLLMs achieve competitive reasoning and calibration using a corpus orders of magnitude smaller than typical LLaMA-scale training, implying substantial reductions in compute and data cost. We will include explicit data-efficiency comparisons, together with the new 10 B-token results, in the updated version.

---

### Author Response · Authors · 2025-12-03
**Final Revision**

We sincerely thank the reviewers for their thorough and constructive feedback. In this final revision, we have carefully addressed all major concerns and substantially strengthened both the empirical and methodological scope of the paper. The main additions and clarifications are summarized below.

(1) Scaling and Data Efficiency.
We added a new large-scale evaluation on a 10B-token subset of the SmolLM corpus (Section 4.6), including full results for factual QA, commonsense reasoning, calibration, and closed-book evaluation. These experiments confirm that all qualitative trends observed at 2.5B tokens persist at scale. We further quantify data efficiency, showing that KLLMs reach comparable contextual factual QA performance with 38–45% fewer training tokens than SLMs. We also clarify that the slightly higher closed-book error at 10B arises from the domain shift of the SmolLM mixture relative to Wikipedia-style closed-book QA data.

(2) Continued Pretraining, Inference-Time Anonymization, and Anonymization-Strength Ablations.
We added an extensive ablation section to the appendix covering:
– Continued pretraining from an SLM checkpoint into a KLLM, showing that the approach is effective both from scratch and under continued training.
– Inference-time anonymization removal, demonstrating that skipping anonymization degrades factual control.
– Anonymization-strength robustness via partial masking (p = 0.0, 0.5, 0.75, 1.0), revealing a smooth, monotonic trade-off between contextual reasoning performance and closed-book factual memorization.
Together, these experiments establish that KLLM behavior is stable across training regimes and that full anonymization yields the strongest separation between reasoning and parametric knowledge.

(3) Closed-Book Knowledge Suppression.
We extended closed-book QA experiments across SmolLM and LLaMA families and added corresponding scaling results at 10B tokens. These results confirm that KLLMs consistently remain near-random in closed-book accuracy, verifying that the model does not recover parametric factual knowledge even at larger scale.

(4) Calibration and Abstention.
We expanded calibration and abstention tuning results across multiple model families and training scales. The new results show that KLLMs consistently achieve higher precision, recall, and F1 than SLMs as uncertainty-aware predictors, strengthening the paper’s claims about faithful confidence estimation and hallucination mitigation.

(5) Transfer from Anonymized to Natural Test Data.
We added a new appendix section evaluating KLLMs on natural (de-anonymized) test inputs under open-book evaluation. Results show that the linguistic abstractions learned under anonymized supervision transfer robustly to real text, with only minor performance changes.

(6) Qualitative Behavioral Analysis.
We added qualitative SLM–KLLM comparisons under open-book evaluation to illustrate hallucination failures of SLMs versus the more grounded or abstent behavior of KLLMs on factual and reasoning benchmarks.

Overall, these additions directly address concerns about scale, robustness, training dynamics, novelty, and empirical completeness. We believe the revision substantially strengthens the paper’s contribution and thank the reviewers again for their invaluable feedback.

---

### Public Comment · ~Roi_Cohen1 · 2026-03-02
**Incorrect statements in the metareview**

As authors on this deanomymized submission, we would like to note that the current metareview consists of fundamentally incorrect statements. In particular, it completely ignores the revisions to the paper made during the discussion period, summarized in the comment dated 3 December. We have appeals to the Programme Chairs, but unfortunately haven't heard anything back. Therefore we would like to clarify it here for the record.

Below we quote the demonstrably incorrect statements made by the metareview, without any references to scientific judgment.

"while the authors promised 10B-token and 8B-parameter runs in rebuttal, we're now past the discussion period with no actual results, which raises red flags about feasibility."

"The promised 10B-token experiments and 8B-parameter models were not delivered during the discussion period"

"None of these promised experiments were actually shown during rebuttal."

"The missing Appendix C undermines credibility."

"Authors defended their choices but provided no new evidence."

"The rebuttal consists largely of promises and justifications rather than new results. Critical experiments (large-scale training, bias evaluation, complete ablations) were repeatedly promised but never materialized during the discussion period, suggesting either infeasibility or insufficient preparation."

 The additional results on 10B-token experiments were added to Section 4.6 "Scaling to 10B Tokens", with results in tables 3-6. We did not train an 8B parameter model: This was suggested by one reviewer but we decided against including this as a scope choice and the final comments and paper does not claim results at 8B+ parameter scale.

Appendix C was added in the revision; We acknowledge that the reference to it in one of the initial reviewer response comments was erroneous;.

The revision also includes additional ablation results in appendices C and D and calibration tuning results (table 7).

---

### Meta-Review · Area_Chair_wS2P · 2026-01-05

**Summary:**

This paper proposes training "knowledgeless" language models via entity anonymization during pretraining, which sounds appealing in principle but falls short in execution and validation. The most fundamental issue is scale. The experiments conducted on 2.5B tokens with models up to 3B parameters is not convicing. The authors claim this is an intentional "controlled" design choice, but that's exactly the problem: if your method only shows benefits at toy scale and you can't demonstrate it works at realistic sizes, the practical value is questionable. Multiple reviewers explicitly requested larger-scale experiments, and while the authors promised 10B-token and 8B-parameter runs in rebuttal, we're now past the discussion period with no actual results, which raises red flags about feasibility.

The 87-90% NER accuracy creates another critical weakness. Yes, it's near SOTA for large-scale tagging, but 13% knowledge leakage is unacceptable for a system claiming to "decouple" knowledge from linguistic competence. The near-random closed-book QA performance does confirm factual suppression, but it also means the model is essentially useless without external context. This is a severe limitation that narrows applicability to RAG-only scenarios. Moreover, anonymization for memory-reasoning disentanglement isn't novel; it's been used in prior work. The contribution here is mainly moving an existing trick to the pretraining stage, which feels incremental rather than conceptually groundbreaking.

Experimental completeness is also lacking. Generation tasks are underexplored, bias mitigation claims remain unsubstantiated (just vague promises of "preliminary internal results" in rebuttal), and calibration improvements are only shown at small scale. The review scores tell the story: one 8, two 4s, and one 2. The concerns about convergence, knowledge leakage, limited scope, and insufficient scale are legitimate and largely unresolved. I recommend rejection. The core idea has merit, but the work needs substantial strengthening.

**Reviewer Concerns:**

Reviewer 4HCZ's concerns:

Partially addressed: The authors provided reasonable explanations for the higher training loss (increased entropy from anonymization) and defended the NER accuracy threshold. However, the core concerns about insufficient scale remain unresolved. The promised 10B-token experiments and 8B-parameter models were not delivered during the discussion period—only vague commitments to include "preliminary results" in a future revision. The data efficiency claim also lacks concrete supporting evidence.

Still outstanding: No actual large-scale experiments; no quantitative data efficiency analysis; method simplicity concerns not addressed (no exploration of alternative anonymization designs).

Reviewer 3Pat's concerns:

Partially addressed: Authors acknowledged all requested ablations and promised to conduct them (continued pretraining, inference-time anonymization removal, alternative NER models, qualitative examples). They also committed to expanding LLaMA coverage across tables.

Still outstanding: None of these promised experiments were actually shown during rebuttal. We have assurances but no data. The scalability question remains empirically unanswered. RAG applicability is discussed conceptually but not demonstrated.

Reviewer TDeV's concerns:

Weakly addressed: Authors claimed their benchmarks already include reasoning tasks (COPA, WSC, CommonsenseQA, etc.) and referenced a non-existent "Appendix C" for anonymized vs. natural data comparison. The reviewer correctly noted in follow-up that Appendix C doesn't exist in the paper.

Still outstanding: Novelty critique not convincingly refuted—anonymization for disentanglement is indeed established in prior work. The missing Appendix C undermines credibility. Generation capability evaluation remains thin. RAG integration is speculative rather than demonstrated.

Reviewer cnzc's concerns:

Not meaningfully addressed: Authors defended their choices but provided no new evidence. The explanation that small corpus size was "intentional" doesn't resolve concerns about generalizability—it reinforces them.

Still outstanding: All major concerns persist: scale inadequacy, knowledge leakage from imperfect NER, limited applicability without context, and complete absence of bias mitigation evidence despite this being a stated motivation in the abstract.

Overall assessment: The rebuttal consists largely of promises and justifications rather than new results. Critical experiments (large-scale training, bias evaluation, complete ablations) were repeatedly promised but never materialized during the discussion period, suggesting either infeasibility or insufficient preparation.

**Reviewer Scores:**

Reviewer 4HCZ (Initial: 4): Likely would remain at 4 or potentially move to 5. The rebuttal provided coherent explanations for training loss patterns and NER reliability, which might satisfy some concerns. However, the lack of delivered large-scale experiments and concrete data efficiency metrics means the core weaknesses remain unaddressed. The reviewer's "marginally below threshold" stance would probably hold steady, though the reasonable engagement might nudge them slightly upward if feeling generous.

Reviewer 3Pat (Initial: 8): Likely would drop to 7, possibly 6. This reviewer was the most supportive but explicitly requested multiple ablations (continued pretraining, inference-time anonymization, alternative NER models, qualitative examples). The authors acknowledged everything but delivered nothing during discussion. A supportive reviewer who asks for specific experiments and gets only promises typically becomes disappointed. The fundamental enthusiasm for the idea would prevent a dramatic drop, but the gap between commitment and execution would matter.

Reviewer TDeV (Initial: 4): Likely would drop to 3. The follow-up comment catching the non-existent Appendix C is telling—it suggests diminishing trust in the authors' claims. The novelty concerns weren't convincingly refuted, and the missing appendix directly contradicts the rebuttal's assertions. A reviewer who starts skeptical and then finds their concerns validated rarely increases their score.

Reviewer cnzc (Initial: 2): Would remain at 2. This reviewer raised fundamental methodological concerns about scale and convergence that the rebuttal didn't resolve—just defended as design choices. The complete absence of bias mitigation evidence, despite being a stated motivation, is particularly damaging. The promises of future experiments would not move a reviewer who already sees the work as "not good enough" without evidence in hand.

---

### Decision · Program_Chairs · 2026-01-26

Reject